# Mating Behavior of *Cyclocephala barrerai* Martínez (Coleoptera: Melolonthidae)

**DOI:** 10.3390/insects16060613

**Published:** 2025-06-10

**Authors:** Abraham Sanchez-Cruz, Patricia Villa-Ayala, Alfredo Jiménez-Pérez

**Affiliations:** Laboratorio de Ecología Química de Insectos, Centro de Desarrollo de Productos Bióticos, Instituto Politécnico Nacional, CEPROBI # 8, San Isidro, Yautepec 62731, Mexico; asanchezc1700@alumno.ipn.mx (A.S.-C.); pvilla@ipn.mx (P.V.-A.)

**Keywords:** mating system, reproductive behavior, ethograms, Markov chains

## Abstract

This comprehensive study delves into the mating behavior of virgin and mated males and females of the beetle *Cyclocephala barrerai*. The mating process in *C. barrerai* is meticulously dissected into three distinct stages, each characterized by specific behaviors: premating, mating, and postmating. During premating, males actively seek and court females, while females initially resist before consenting to mate. The mating involves males mounting and continuously stimulating females while females display specific head movements. Postmating, males guard the females to prevent them from mating with other males. This study uncovers clear differences in the behaviors observed during the premating and postmating stages between virgin and once-mated organisms. However, previous matings did not influence the time invested in searching for, courting, or mating with a female. This work provides a comprehensive understanding of mating in *C. barrerai* and serves as a crucial point of comparison for future research into its physiology, morphology, and chemical communication.

## 1. Introduction

Mating is the sexual encounter between a female and a male to transfer the male’s gametes (spermatozoa) to the female and fertilize her eggs [1]. Mating is mediated by specific actions [2], identifying recognizable patterns that generate a well-defined sequence that can be analyzed and expressed in a first-order Markov chain, for example, the description of the mating behavior patterns of fruit flies [3].

Studies of insect mating behavior have identified four core elements—mate choice [4], courtship [5], genital coupling [6], and mate-guarding [7]—which reveal how reproductive strategies evolve across species [8]. This knowledge can inform both pest management programs [9] and conservation efforts [10,11]. Factors such as age [12], photoperiod [13], and a female’s prior mating history [14,15] can alter the typical behavioral sequence: young, virgin insects at the start of the season behave differently than older, once-mated individuals under mid-season conditions. Insects learn from experience, so innate behaviors shown in a first, virgin encounter may change in subsequent matings [16].

Consequently, once-mated individuals often exhibit significant behavioral changes in subsequent matings. In the butterfly *Heliconius erato phyllis* (Fabricius), for example, once-mated males display fewer precopulatory behaviors than virgins: they expose the androconial glands less frequently, perform fewer abdominal movements, and show a longer average wing flutter duration during courtship [17]. In the moth *Achroia grisella* (Fabricius), copulation with once-mated females lasts longer than with virgins, as males invest more time transferring larger quantities of sperm to enhance their chances in sperm competition [18]. Similarly, in *Ostrinia nubilalis* (Hübner), once-mated males are more likely to be chosen by females compared to virgins. They show greater mate-searching activity, and it has been suggested that changes in their pheromone profile may increase their attractiveness [19].

Research on Melolonthidae family species [20] has clarified key aspects of adult reproductive biology. These beetles spend most of their lives underground, emerging only at the rainy season’s onset—their “activity period”—to feed and mate [21,22]. Most species are crepuscular or nocturnal (e.g., *Phyllophaga cuyaba* [22], *Liogenys fusca* [23]), while *Macrodactylus* (subfamily Melolonthinae) species are active by day [24]. At mating onset, males locate females by tracking a sex pheromone plume. Females “call” by exposing their genital chamber and emitting volatile compounds [22,25], or they may rely on contact pheromones on the cuticle [26,27,28]. Although courtship rituals vary, males generally attempt to insert the aedeagus and transfer sperm [29,30]. Females may resist by walking away, rolling, or pushing the male off, or they may remain immobile [23,27,31,32].

The genus *Cyclocephala* Dejean is highly diverse and distributed throughout the Americas. More than 300 species have been recorded in a wide variety of habitats, displaying different biological adaptations [33,34]. Only six species of *Cyclocephala* have been described in detail regarding their mating behavior [33,34]. The species *C. melanocephala* (Fabricius) [35], *C. putrida* (Burmeister) [36], *C. distincta* (Burmeister) [37], and *C. lunulata* (Burmeister) [38,39] are distributed in tropical areas. Their adults have extended mating activity schedules, lasting more than four hours, except for two hours for *C. distincta* [37]. They combine feeding with mating, so it is common to observe them feeding and mating on inflorescences or soft-barked fruits. In contrast, *C. borealis* (Arrow) [40,41] and *C. immaculata* (Bland) [41] have limited mating activity schedules lasting less than two hours. Their adults do not feed and are focused on mating. In *C. distincta* [37] and *C. immaculata* [41], sex pheromones involved in mate finding and recognition are detected. The study of the behavior of these six *Cyclocephala* species has been recorded in organisms in the field or ones that have been collected from the field and brought to the laboratory, and their life history and physiological characteristics are unknown.

*Cyclocephala barrerai* Martínez is distributed in central Mexico and inhabits grasslands and turfs, coexisting with other Melolonthidae species within the so-called “white grub complex” [42,43,44]. The larvae of *C. barrerai* are facultative feeders that consume organic matter in the soil and plant roots (Sanchez-Cruz et al. [45] (in press)). These larvae play an ecologically important role by feeding on organic matter; they contribute to nutrient cycling in the soil, and their burrowing activity promotes soil aeration by creating tunnels [46]. Although *C. barrerai* larvae feed on the roots of Poaceae species [42,47], they have been associated with damage to ornamental grass crops; however, the extent of this damage has not yet been accurately quantified. There is morphological and physiological evidence that mating is a critical stage in the life cycle of *C. barrerai* and that it has led to several morpho-anatomical adaptive mechanisms [48,49]. The study of the morpho-anatomical adaptations involved in the reproduction of *C. barrerai* has established this species as a model for mating studies within the Melolonthidae family [28,48,49].

Several morphological and physiological aspects of *C. barrerai* reproduction are currently known. *Cyclocephala barrerai* adults’ gonads (oocytes and sperm) mature after the adult emerges from the pupa, before any sexual behavior is initiated [49]. Females of *C. barrerai* have developed a genital chamber characterized by specialized musculature, which has been hypothesized to be a mechanism to control or stimulate spermatophore transfer by the male. In addition, both sexes have ultrastructures in the sclerotized parts of the reproductive apparatus that could have mechanical or chemical receptor functions [49]. Male antennae of *C. barrerai* have undergone a selection process that has resulted in disproportionately larger structures, with a larger number of sensilla placodea type IV [48,49]. These sensilla, present on females’ and males’ antennae, are responsible for chemoreception, i.e., they are the main olfactory structure detecting plant volatiles and cuticle-related hydrocarbons from conspecifics [48].

Information about the mating behavior of *C. barrerai* is limited. Under natural conditions, adults of *C. barrerai* are nocturnal and have not been observed feeding [42,50]. Males walk on the grass, searching for females between the grass and the ground. Although males have been recorded positioning themselves over females to copulate, the position of the females (between the grass and the ground) did not allow courtship, mating, postmating, or specific sexual behaviors to be clearly observed [50].

There are likely volatile organic compounds involved in the mate searching for *C. barrerai*. Adults of *C. barrerai* captured in the field maintained similar reproductive activity when transferred to the laboratory, and males responded positively when approaching the source of bacterial volatiles in the genital chamber of females [51]. In contrast, females did not approach the odor source. However, in electroantennography tests, the antennae of both sexes detect bacterial volatiles [40]. Although specific sexual behaviors were recorded during the study, it was impossible to relate these directly to mating due to the lack of detailed studies describing such behaviors.

The mating behavior of *C. barrerai* has not been studied in detail, which has prevented the identification of direct relationships between the morphological and physiological characters of the species. A detailed analysis of mating behavior is essential to improve our knowledge of the reproductive biology of *C. barrerai*. However, field observations are hampered by limited visibility and access to the behavior of organisms in their natural habitat. When working with populations of unknown characteristics, details of mating behavior may be lost [52], especially when working with mated organisms because the first mating may vary in terms of behavioral acts and duration [53].

Studying the mating behavior of *C. barrerai* under controlled laboratory conditions is a crucial first step toward a detailed characterization of its mating behavior and a broader understanding of animal reproductive strategies [54,55]. However, laboratory studies cannot explain animal behavior definitively [44]. Still, in insects, they have served to identify important details of mating, for example, in *Drosophila* spp. [56] or *Culex tarsalis* (Coquillet) [57]. Information on mating behavior and the factors that modify it has been the foundation for establishing various management programs, such as the sterile insect technique [58] or mating disruption [59].

The general objective of this study is to characterize, under controlled laboratory conditions, the mating behavior of *C. barrerai* adults. The specific objectives are (1) to characterize mating behavior as a function of sex (female and male) and (2) to record behavioral changes between virgin and once-mated organisms. The results reported in this paper will contribute to our understanding of innate behavioral responses in males and females during their first sexual encounter with conspecifics, in the absence of external variables, and how these instinctive behaviors are modified in individuals with prior mating experience.

## 2. Materials and Methods

### 2.1. Insects

The organisms used were reared under the same laboratory conditions as reported by Sanchez-Cruz et al. [49]. In summary, the egg to prepupal stages were reared individually, with one organism in a 500 mL plastic container (#16, Reyma, León de los Aldama, Mexico) with a food substrate made from leaves, wood, soil, and grassroots. All instars developed buried in the substrate. Pupae were then individualized in 50 mL containers (Molde 501, Reyma, Mexico) with substrate from the previous container and monitored until emergence.

The weight of newly emerged adults was measured using an analytical balance (E02140, Ohaus Explorer, Ohaus, Switzerland) to ensure that the adults used in this study were close to the average of the field populations. The tarsal nail of the first pair of legs was used to identify their sex. During the first 24 h after emergence, the adults were in the pupae containers. After that, they were transferred to new 500 mL plastic rearing containers and reburied in the substrate. All rearing was under controlled conditions of 24 ± 2 °C, 60 ± 10% RH (relative humidity), with an inverted photoperiod of 12IL:12ID similar to its natural habitat.

### 2.2. Study of the Age at the Beginning of the First Period of Sexual Activity and Its Duration

Adult *C. barrerai* (*n* = 40) were kept individually in their rearing containers and monitored daily to record the duration and age at the first period of activity [21,22]. Age was defined as the number of days since adult emergence. The duration of the mating activity period corresponded to the number of minutes from digging up from the substrate until reburial.

### 2.3. Recording of Mating Behavior

The experiments were carried out in a cylinder-shaped bioassay arena made of acrylic material (10.55 cm high × 20 cm diameter) on top of a 20 cm high by 20 cm wide by 20 cm long acrylic box (Figure 1). The room was kept at 24 °C, 60% RH, and illuminated with a red light bulb (Philips, Shenzhen, China) at 002 lux, as shown by a digital lux meter (Her-410, Steren, Mexico City, Mexico) (Figure 1). Red light does not alter the period and timing of adult *C. barrerai* activity, nor does it alter mate-searching behavior [51]. The behavior of the organisms was filmed using four webcams (pro C920, Logitech, Newark, CA, USA) placed around the bioassay arena (one above the arena, one below, and two on the sides) (Figure 1). The computer that recorded the video from the cameras and the observer were in a neighboring room to prevent light and odor from interfering with the behavior of the insects.

It is unknown what triggers the mating activity in *Cyclocephala* species, but we know it only occurs in darkness [40,41]. So, we checked our lab colony every day during the inverted dark period and collected the adults on the surface of the substrate (activity period). These insects were transferred in the dark to the bioassay room in transparent 50 mL containers, each adult separately.

A female was placed in the arena and left alone for 10 s. Then, a virgin male was introduced, and they were observed continuously during mating and least 10 min after that. After mating, the female and male were returned to their breeding containers. Each female’s container was checked for eggs seven days later.

Two different bioassays were performed:(1)Virgin adults (*n* = 26 couples): adults who were in their first period of mating activity, 25 to 27 days post-emergence, and had no earlier mating experience.(2)Once-mated adults (*n* = 15 couples): these adults came from the previous bioassay and were in their second period of activity. The age of these insects ranged from 34 to 36 days post-emergence.

The videos were analyzed using Behavioral Observation Research Interactive Software (BORIS) version 8.13 [60] to identify behavioral acts and transitions. The duration of behavior at each sexual stage was also recorded. For clarity, the sexual behaviors of each sex were broken down into three phases:(1)Premating—This phase begins when the insects emerge from the soil and are allocated to the bioassay arena, where they search for and court a potential mating partner [61]. Blue color was used in the ethogram to represent this phase.(2)Mating—This phase involves the union of the reproductive apparatus, insertion of the male’s aedeagus into the female, and ejaculation by the male [62]. Red color was used in the ethogram to represent this phase.(3)Postmating—This phase begins after the male transfers an ejaculate and withdraws his aedeagus [30]. Gray color was used in the ethogram to represent this phase.

### 2.4. Statistical Analysis

Ordered behavior transition matrices (Markov chains) were used to describe the first-order behavior transitions [63]. Behavioral acts were included in Markov tables if they occurred in at least 10% of the repetitions. A transition was considered significant if the observed frequency exceeded the specified frequency. The χ^2^ value and the percentage of transitions were calculated for significant transitions. Ethograms of significant transitions were generated using Graphviz software (version 12.2.1) [64,65]. The sketches were generated using Krita software version 4.4.2.

The comparison of the duration of the stages of sexual behaviors between virgin and once-mated organisms was performed with a *t*-test. The behavioral transition proportions between virgin and once-mated organisms were compared using a Z-test. These analyses were performed using SigmaPlot software (version 14.0.0.124, Systat Software Inc., Erkrath, Germany) with a 0.05 rejection probability.

## 3. Results

### 3.1. Age at the Beginning of the First Period of Sexual Activity and Its Duration

Lab-reared adults were first active at 25.4 ± 1.34 (Mean ± ESM) days of age. Activity began one hour after the lights were turned off (total darkness) and lasted 59.87 ± 6.8 (mean ± ESM) min. At the end of this time, adults reburied. When male *C. barrerai* failed to mate, they were active every two to three days. Female activity was highly variable and ranged from four to eight days.

### 3.2. Mating Behavior of Virgin and Once-Mated Adult Cyclocephala barrerai

In total, 26 virgin couples and 15 once-mated couples of *C. barrerai* adults tested in the bioassays mated. Mated females lay fertilized eggs. Appendix A show the mating behavior of virgin insects, while Appendix A show the mating behavior of once-mated organisms. The link to the videos can be found in the Appendix A.

#### 3.2.1. Male Mating Behavior

A total of 13 behavioral acts comprised the male sexual activity repertoire (Table 1). Eight acts were performed during the premating phase, two during mating, and two more during the postmating phase. Virgin and once-mated males performed 11 and 13 behavioral acts, respectively.

Table 2 presents the significant first-order behavior transitions (Σχ^2^ = 1646.4, df = 90, *p* < 0.05) in the mating behavior of virgin *C. barrerai* males. At the onset of premating, virgin males walk (MW) frantically around the periphery of the arena (Figure 2, Appendix A). They keep the lamellae of the antennae open and facing forward. Thirty-four percent of virgin males make alternating antennal contraction and extension movements while opening and closing the lamellae (MHAM) (Appendix A). However, during premating of virgin males, only 2% of the time did virgin males who performed the MHAM act return to MW behavior. During the remaining time, after exhibiting the MHAM act, males walked directly towards the female (MWDF), completing the transition sequence MHAM → MTDF → MWDF → MATFB (Figure 2, Appendix A). This explains that 50% of virgin males that started with the behavioral act MW presented the sequence MW → MTDF → MWDF → MATFB (Figure 2). In only one case did the male walk around the bioassay arena where the female had passed until he met her (Appendix A), and on one occasion, the female approached the male (Appendix A).

In all experiments, a male walked to the female and antennated her with his antennae on the abdomen (Appendix A), pygidium (Appendix A), or thorax (Appendix A) (MATFB) (Figure 2). The MATFB act marks the beginning of the courtship. At this stage, all males displayed the MHFAR behavioral act (Figure 2), positioning themselves behind the female and mounting her, hooking onto the female’s elytra by the tarsal claws of their first pair of legs, and moving the abdomen to a position parallel to the female’s pygidium while frantically rubbing their antennae on the female’s elytra (Appendix A).

If the male fails to grasp the female firmly, she throws him off, and he walks back to her (MW) (Figure 2, Appendix A). The female threw 34% of the males during MHFAR behavior, but all these males managed to copulate with the female on a second attempt (Figure 2).

When the male succeeds in aligning his body with the female, he stops swinging, extends the aedeagus (MEE act), and pushes his abdomen forward until it is inserted into the female’s pygidium (Figure 2, Appendix A). In these experiments, the female failed to displace any male that achieved the MEE behavioral act (Figure 2).

The MEE → MIE transition marks the end of the premating stage and the beginning of the mating stage (Figure 2). During mating (MM), the male remained mounted on the female and rhythmically rubbed his antennae against her elytra (Figure 2, Appendix A). Less than 10% of males rubbed one of the third pair of legs on the female’s abdomen. At the end of mating (MM), the male disengages the aedeagus, retracting it into his abdomen, and guards her (MG) (Figure 2). The MG is the beginning of postmating because, although the male is still on top of the female, holding her with the tarsal claws of the first pair of legs, the aedeagus separates from the pygidium of the female (the aedeagus is not inserted), the male does not present antennal rubbing, and his abdomen is not entirely aligned with the female (Appendix A). All virgin *C. barrerai* males showed guarding, and only 30% released the female to continue walking in the bioassay arena (MWP).

Significant first-order behavior transitions (Σχ^2^ = 1093.5, df = 120, *p* < 0.05) in the mating behavior of the once-mated males of *C. barrerai* are presented in Table 3. During the searching behavior of once-mated males in the premating phase, 53% walked (MW) when placed in the bioassay arena (Figure 3, Appendix A). In comparison, the remaining 47% stayed still (MQ, Appendix A) and subsequently showed the transition MQ → MW or MQ → MHAM (Figure 3). Seventy-three percent of the once-mated males exhibited directed movements to locate the female, exhibiting the transition MTDF → MWDF → MATFB (Figure 3, Appendix A). The remaining 27% continued walking (MW) until encountering the female (Appendix A), undergoing the transition MW → MATFB (Figure 3). Notably, the MW → MATFB transition was only observed in once-mated males (Table 3).

The behavioral sequence of the once-mated males during courtship was MATFB → MHFAR → MEE (Figure 3). All once-mated males touched the female’s body with their antennae before initiating courtship (Appendix A). Then, they mounted the female, avoiding being thrown off by her. Finally, they aligned her body and inserted the aedeagus (Figure 3). During mating, the once-mated males presented the sequence MIE → MM (Figure 3).

At the end of mating, all once-mated males transitioned from MM to MG (Figure 3). While all once-mated males exhibited MG behavior, only 20% retained this behavior until the end of the bioassay (Figure 3, Appendix A). Forty percent of the once-mated males showed the MG →MWP transition (Figure 3). The remaining 40% made the MG → MQP transition (Appendix A). In this case, the males guarded and released the female and stayed still until the end of the bioassay.

The percentage of virgin (84.6%) and once-mated (53%) males that walked at the beginning of the experiment was not significantly different (z = 1.8, *p* = 0.061). The percentage of virgin (34%) and once-mated (26%) males that exhibited MHAM behavior (z = 0.18, *p* = 0.85) and the percentage of males thrown by females during courtship was not significantly different (z = 0.18, *p* = 0.85).

Virgin males spent, on average, significantly less time (mean ± ESM, 387.5 ± 46.8 s) guarding the female than once-mated males (219 ± 52 s) (t = 0.8, df = 39, *p* = 0.02). No other significant differences in searching time, MHFAR behavior, MEE behavior, courtship, MM behavior, and total time of mating duration were found between virgin and once-mated males (SM1).

#### 3.2.2. Female Mating Behavior

The female sexual activity repertoire comprised 11 behavioral acts (Table 4). Eight behavioral acts correspond to premating, two to mating, and one to postmating. Virgin and once-mated females performed 11 and 8 behavioral acts, respectively.

The significant first-order behavior transitions (Σχ^2^ = 1116.24, df = 81, *p* < 0.05) in the mating behavior of virgin *C. barrerai* females are shown in Table 5. During premating, 80% of all virgin females in FW moved around the periphery of the arena (Figure 4, Appendix A). Fifty-four percent of them maintained this behavior until being intercepted by the male (Figure 4, Appendix A). The remaining 46% of females performed three transitions: (1) the females stopped walking and exhibited antennal movements, FW → FHMCA transition (Figure 4, Appendix A); (2) the females moved in the opposite direction of the male, FW → FMA transition (Figure 4, Appendix A); and (3) the females moved in the direction of the male, FW → FTDM → FWDM transition (Figure 4, Appendix A).

Forty percent of virgin females performed the FQ behavioral act (Figure 4). All females in FQ performed antennal movements before moving, showing the FQ → FHMCA transition (Figure 4). After the antennal movement, the females moved in the opposite direction to the male, FHMCA → FMA transition (Figure 4), or turned towards the male to walk towards him, FHMCA → FTDM → FWDM transition (Figure 4).

Although some females approached the male, fewer than 10% of them made contact (Appendix A). The male always reached the female and made contact (Appendix A). When the female and the male had contact, the female stood still momentarily, so the male touched her with his antennae in the FMM act (Figure 4, Appendix A). The FMM behavior occurred in all bioassays and marked the end of the search phase and the beginning of the courtship phase for females.

During courtship, all virgin females exhibited the FWK behavioral act (Figure 4), walking and energetically kicking the male (Appendix A). Thirty-four percent of the females succeeded in detaching from the male (FWK → FW transition, Figure 4), whereby the females walked back to the periphery of the arena (Appendix A). When the female did not displace the male, the male inserted the aedeagus (FWK → FM transition). This is the commencement of mating (Figure 4).

At the onset of mating, virgin females strolled through the bioassay arena with the males on their backs (FM, Figure 4, Appendix A). The second mating phase occurred with the FM → FQHM transition (Figure 4). During FQHM, the female abruptly halted her walking activity and stood still with the male on her back. She moved her head back and forth while contracting and extending her antennae (Appendix A). The FQHM behavior was observed in all bioassays (Figure 4).

The postmating phase follows the end of the FQHM behavior. During postmating, all virgin females walked the periphery of the arena, regardless of whether they were carrying the male. This behavior continued until the end of the bioassay (see Figure 4, Appendix A).

The significant first-order behavior transitions (Σχ^2^ = 506.9, df = 49, *p* < 0.05) in the mating behavior of once-mated females of *C. barrerai* are shown in Table 6. When placed in the bioassay arena, all once-mated females of *C. barrerai* showed the behavioral act FW (Figure 5, Appendix A). During the search phase, 53.3% of the once-mated females maintained the FW behavior until being intercepted by the male (Appendix A) (Figure 5). The remaining 46.6% presented the transition FW → FTDM → FWDM (Figure 5) when encountering the male (Figure 5, Appendix A).

All once-mated females displayed FMM behavior when intercepted by the male (Figure 5). During courtship, all once-mated females showed the FMM → FWK transition (Appendix A). Forty percent of the once-mated females succeeded in pushing the male away by the FWK act. When the male was removed, the once-mated females presented the FW act again (Figure 5). When the once-mated females successfully pushed the male away, mating occurred in subsequent attempts.

During mating, all once-mated females showed the FM → FQHM transition (Figure 5, Appendix A). At the end of mating, all once-mated females displayed the behavioral act FWP, regardless of whether the male was on their backs (Figure 5, Appendix A).

Virgin and once-mated females spent similar amounts of time performing the FWK (t = 0.83, df = 39, *p* = 0.4), FM (t = 1.5, df = 39, *p* = 0.14), and FQHAM (t = 0.67, df = 39, *p* = 0.5) behavioral acts. No other significant differences in FWK, FM, and FQHM were found between virgin and once-mated females (SM2).

## 4. Discussion

*Cyclocephala barrerai* adults are active at night and focused on reproduction, as reported in the wild [42,50]. Compared to other *Cyclocephala* species, the activity period of *C. barrerai* of one hour, is short, regardless of whether it is their first or second mating. For example, in South American species, adults of *C. melanocephala* are active for 8 h [35] and *C. putrida* for 10 h [36], while *C. distincta* is active for 2 h, combining mating and feeding [37]. The adults of *C. barrerai* do not feed. The males walk through the grass looking for the female, and the females stay at ground level, waiting to be intercepted by a male. After mating, they return into the ground. In North American species, activity periods are more limited. For example, *C. immaculata* and *C. borealis* are active at night for 2 and 4 h, respectively. They do not feed during the adult stage, and their activity is focused on mating [41]. *Cyclocephala barrerai* has the shortest copulation time compared to other *Cyclocephala* species: 10 ± 4 min in *C. melanocephala* [35] and 7.5 ± 1.8 min for *C. putrida* [36]. This time difference is likely because mating was not interfered with by any other factors in our experiments.

In adult insects, energy resources must be efficiently managed for all physiological and metabolic processes to occur correctly. Reproduction requires gonad maturation, pheromone development, and mating [66]. In the case of *C. barrerai*, grasslands at the end of the rainy season fail to have enough energy resources, and foraging may be counterproductive. This may explain why adults are only active after 25 days of their eclosion, when they are sexually mature [49].

Our laboratory bioassays demonstrated that *C. barrerai* males typically search for and find the female during premating; this behavior is consistent with field observations [50]. We are confident that this behavior also occurs in their natural habitat, though it may not yet be documented due to the female’s ability to move among the grass. In *C. distinta*, the males chase the females for mating [27]. In other Melolonthidae species, it is also the male that actively seeks and courts the female: this is the case in *Phyllophaga vetula* (Horn) [67], *Anomala testaceipennis* (Blanchard) [26], and *Liogenys suturalis* (Blanchard) [32].

Melolonthidae males have been observed to walk with the antennae extended and lamellae open when searching for a female. This characteristic was also recorded in *C. barrerai* during this study. In mating bioassays, *C. barrerai* males turned and walked in a directed manner toward the female to mate with her. While visual signals may have triggered this behavior, pheromones are also a likely factor, as the males did not immediately approach the female after being placed in the arena. Instead, they exhibited a characteristic antennal movement (MHAM) before moving towards the female. In insects, this behavior is usually associated with detecting attractive odors, and males orient themselves toward the source of stimuli through antennal movements [68]. This pattern has been reported in other Coleoptera, including *Anomala orientalis* (Waterhouse) [69] and *Thanasimus dubius* (Fabricius) [70].

*Cyclocephala barrerai* females have small antennae, while males have disproportionately large antennae, with many specialized chemoreceptive sensilla capable of detecting hydrocarbons [48]. These hydrocarbons are part of the cuticle of adults. It is, therefore, reasonable to propose that pheromones mediate communication between males and females. Field observations of *C. barrerai* mating indicate the presence of a volatile substance that mediates male searching. This is supported by the apparent invisibility of the females when buried in the soil [50]. Volatile substances from the host plant [71,72] or pheromones from the females [41,73] mediate mate searching in *Cyclocephala* species.

In our study of the mating behavior of *C. barrerai*, as with field observations of other species in the genus *Cyclocephala*, no female calling has been recorded. Calling has been documented primarily in species of the subfamily Melolonthinae, such as *Phyllophaga obsoleta* (Blanchard) and *Ph. vetula* [67,74], and the subfamily Rutelinae, in *Anomala inconstans* (Burmeister), where females stridulate extending the abdomen and rubbing it with their legs [75]. So, the “possible pheromone” may be a passively disseminated substance like the volatiles of the resident bacteria of the genital chamber [51] or a trail pheromone composed of cuticular hydrocarbons [28].

A turning point during premating is when the *C. barrerai* male touches the female’s body with his antennae, triggering the courtship behavior. This behavior was performed in all mating of virgin and once-mated males. This behavior has been recorded in *C. melanocephala* [35], *C*. *putrida* [36], and *C. distincta* [27]. In *C. distincta*, males can discriminate between females and males by touching their conspecifics with their antennae, suggesting a chemical recognition [27].

Once-mated *C. barrerai* adults show simplified premating behavior sequences. Females and males are more direct in finding a mate, staying still, and moving only their antennae before approaching a mate. This behavior is consistent with that reported for the *Drosophila* species [53]. This cautions behavior reduces the time couples are exposed to and minimizes the probability of being detected by natural enemies. Previous mating experience likely helps both males and females identify potential partners more efficiently, allowing them to engage in more direct courtship behaviors [76]. In *C. barrerai*, it is plausible that both sexes rely on visual cues or detect sex pheromones more effectively. Based on previous studies in lepidoptera [17,19], it is also possible that physiological changes in adult *C. barrerai* influence mate searching and selection. However, the underlying mechanisms remain largely unknown in Melolonthidae.

The fighting behavior of *C. barrerai* females during courtship is not different between virgin and once-mated females. The females of *C. barrerai* show a constant fighting behavior, in contrast to other species of the genus *Cyclocephala,* where only 14 to 28% of females show this behavior [35,36]. In *Macrodactylus* spp., females fight the male by kicking him [5]. Females of *L. fusca* swing sideways to pull males away [23], while *A. testaceipennis* females roll on the ground to reject the male’s attempts [26]. In *C. borealis*, *C. lurida* [41], and *C. putrida* [36], females walk away from the male during courtship. In the family Chrysomelidae, females of *Macrohaltica jamaicensis* (Fabricius) showed similar fighting behavior to females of *C. barrerai* in a laboratory bioassay [77]. These females’ fighting behavior may be part of their female cryptic choice mechanisms. Eberhard [78] explains that during courtship, the male must overcome the female’s selection mechanisms until the female allows for the introduction of his reproductive apparatus and the transfer of spermatozoa occurs.

During courtship and copulation, the male of *C. barrerai* constantly rubs his antennae against the female’s elytra. This rubbing is maintained during copulation. However, such behavior has been observed in *L. fusca*, and it is related to the detection of a pheromone in the female’s cuticle [23]. During courtship, beetles’ females use several mechanisms to evaluate the male’s sperm quality. Their courtships involve males stimulating the female by rhythmically rubbing a specific structure on their bodies. *Diabrotica undecimpunctata howardi* (Barber) males that rub their antennae more intensely on the female’s body have a higher acceptance percentage in a shorter time than those that do not express such behavior. Males that had their antennae removed were never accepted by females for mating [79].

*Cyclocephla barrerai* clearly shows a relationship between the positive allometric development of male antennae and the specialization and the thickening of the beak-like ultrastructure of the tarsal nails of the first pair of legs [49]. During courtship behaviors, the male uses the nails of the first pair of legs to latch onto the female and avoid being displaced while stimulating her with his antennae and inserting the aedeagus. In insects, specialized structures for grasping the female are considered courtship devices [78]. An inadequate male (small or old) may not overcome female resistance due to ineffective antennal stimulation or attachment. In this study, we used males of uniform size and age, which allowed all to be successful in courtship. In general, the *C. barrerai* organisms used our study were young and healthy.

The head and antennae movement recorded in *C. barrerai* females during copulation has never been reported in any other species of Melolonthidae. Eberhard [80] confirms that when females fully accept the male, they redouble their external movement to produce an internal abdominal pumping or movement to favor sperm transfer. In the study of *Ma. Jamaicensis*, they reported that external behaviors during copulation are related to the internal interaction of the reproductive apparatus and the formation of spermatophores [77]. A similar process likely occurs during copulation in *C. barrerai*.

The genital chamber of *C. barrerai* females is built of muscle tissue and has undergone selection pressure, meaning that larger organisms develop disproportionately larger genital structures [49]. During copulation, the male’s endophallus passes through and is enveloped by the genital chamber. This interaction gives the female control over sperm transfer by preventing or stimulating spermatophore formation. This control is key to avoiding mating with unfit individuals and reduce the risk of hybridization [81,82].

The absence of differences between virgin and once-mated individuals of *C. barrerai* during courtship and copulation suggests two possible hypotheses: (1) The behaviors recorded in this study, together with the morphological and physiological traits previously described in *C. barrerai* adults [48,49], may constitute a necessary mechanism to ensure optimal insemination [4] and are likely part of a sexual selection process [5,83,84]. (2) A second mating may not be sufficient to induce significant physiological changes in the insect that could affect its behavior. This second hypothesis could explain why, as observed in some lepidopteran species, a single spermatophore or an insufficient postmating interval after the first copulation may not be enough to critically alter the adult physiology and, consequently, its insemination behavior [17,19]. Further detailed studies on other Melolonthidae species are needed to clarify this issue. Postmating guarding is a definitive characteristic of *C. barrerai* males. In insects, guarding is a postcopulatory behavior used by males to prevent females from copulating with other males. This behavior reduces sperm competition and ensures or increases the likelihood that eggs will be fertilized with their sperm [7]. *Cyclocephala barrerai* males exhibit a “mate grip” type of guarding, meaning they remain on top of the female in the copulating position, even though their reproductive organs are separated [7].

The present study definitively shows that 80% of *C. barrerai* virgin males guarded the female until the end of the bioassay while the female walked in the bioassay arena. This behavior may not be so extended in their natural habitat, as males release the female when she buries herself again, as described in *A. orientalis* [69]. In contrast, 80% of the once-mated males of *C. barrerai* did not maintain the guarding behavior until the end of the experiment, instead opting for other behaviors associated with the search for shelter or another mate. This behavior change is likely due to their previous energy expenditure during the first and second mating, leading them to prioritize different behaviors [4].

## 5. Conclusions

Adult *C. barrerai* are reproductively active at about 25 days of age, when they are sexually mature. The reproductive period is brief, regardless of whether they mate or not. Males engage in active mate searching, and their behavior differs depending on prior mating experience, suggesting a chemical recognition mechanism that guides female detection and initiates courtship. While courtship and copulation behaviors are broadly similar between virgin and once-mated individuals, variations in guarding behavior and female resistance highlight the dynamic interactions during mating. During premating, virgin *C. barrerai* females are more behaviorally active than once-mated ones, often exhibiting resistance that males must overcome to copulate. During copulation, females display two distinct phases associated with internal reproductive interactions, after which mating behavior concludes.

## Figures and Tables

**Figure 1 insects-16-00613-f001:**
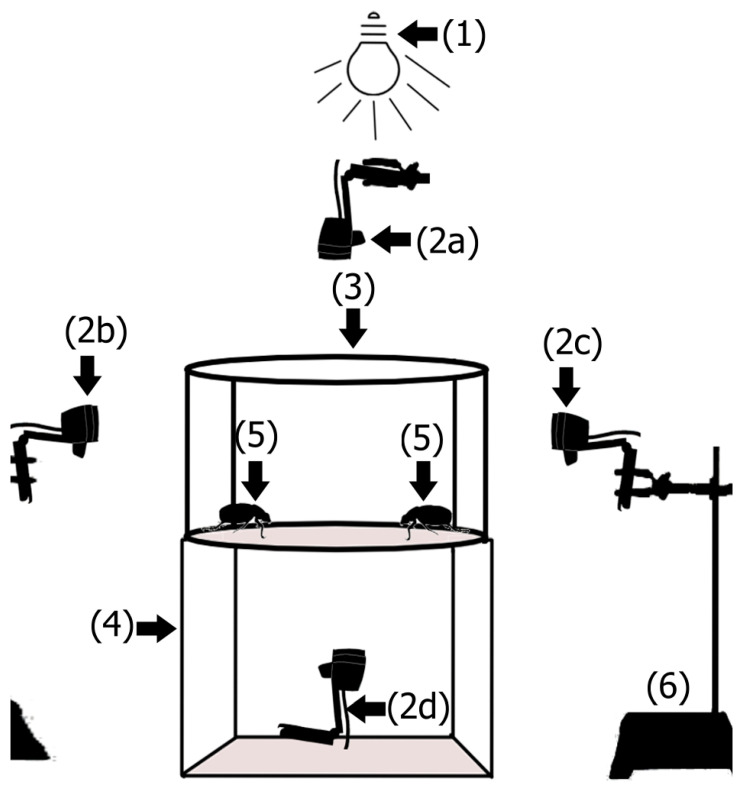
Bioassays of the mating behavior of *Cyclocephala barrerai*. Mating bioassays were conducted under red light illumination (1). The assays were recorded using a top-mounted camera (2a), two lateral cameras—one on the left side (2b) and one on the right side (2c)—and a bottom-mounted camera (2d). The circular bioassay arena (3) was suspended above a cubic acrylic box (4). A male and a female adult of *C. barrerai* were introduced into the arena to allow for mating observations (5). Universal support to hold cameras (6).

**Figure 2 insects-16-00613-f002:**
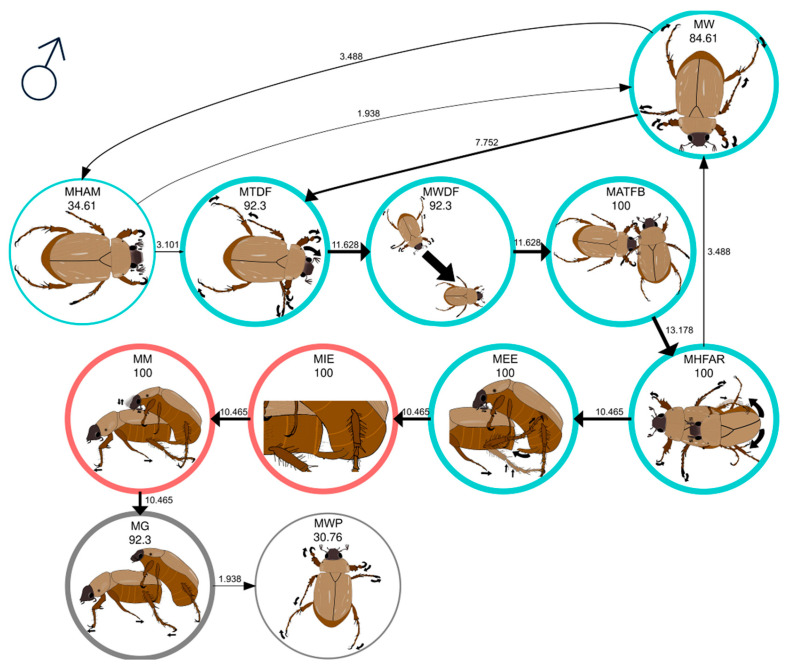
Mating behavior of virgin *Cyclocephala barrerai* males. The thickness of the line is proportional to the percentage value of transitions; this value is to the right or above the line. The line thickness of the circles is proportional to the percentage of the behavior; the percentage value is below the behavior name. The color of the circles indicates the phase of the mating process: blue = premating; red = mating; and gray = postmating.

**Figure 3 insects-16-00613-f003:**
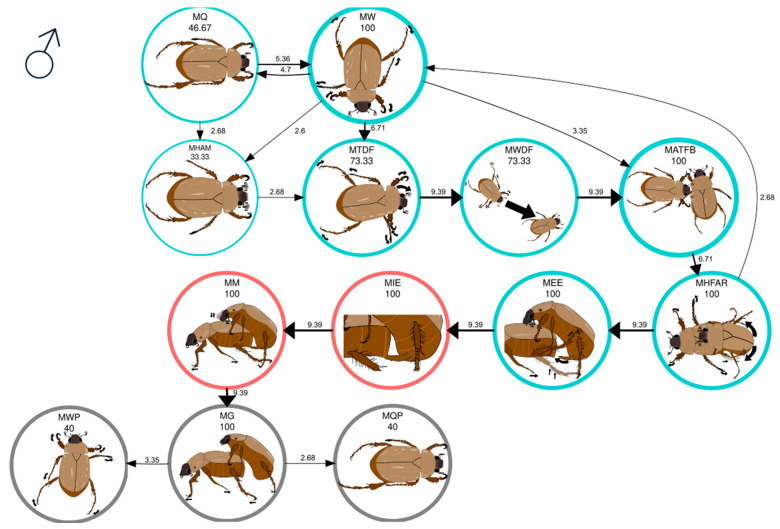
Mating behavior of once-mated *Cyclocephala barrerai* males. The thickness of the line is proportional to the percentage value of transitions; this value is to the right or above the line. The line thickness of the circles is proportional to the percentage of the behavior; the percentage value is below the behavior name. The circle’s color represents the phases of the mating process: blue = premating; red = mating; and gray = postmating.

**Figure 4 insects-16-00613-f004:**
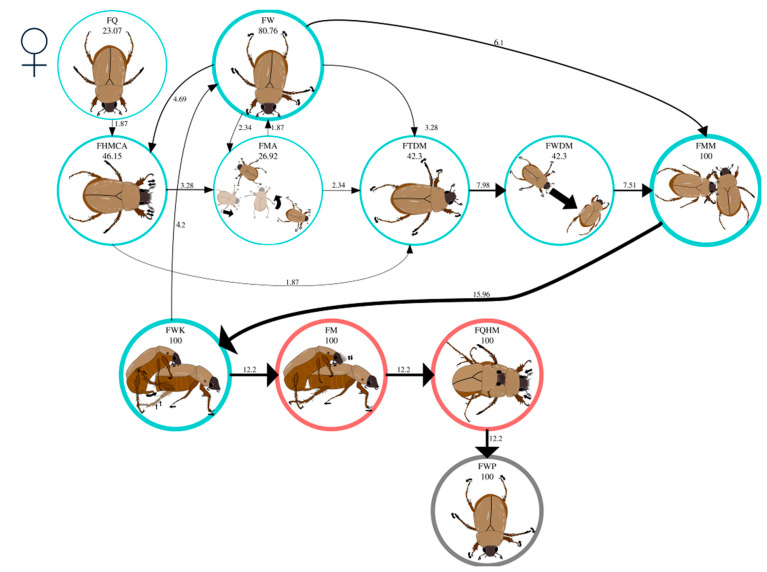
Mating behavior of virgin *Cyclocephala barrerai* females. The thickness of the line is proportional to the percentage value of transitions; this value is to the right or above the line. The line thickness of the circles is proportional to the percentage of the behavior; the percentage value is below the behavior name. The circle’s color represents the phases of the mating process: blue = premating; red = mating; and gray = postmating.

**Figure 5 insects-16-00613-f005:**
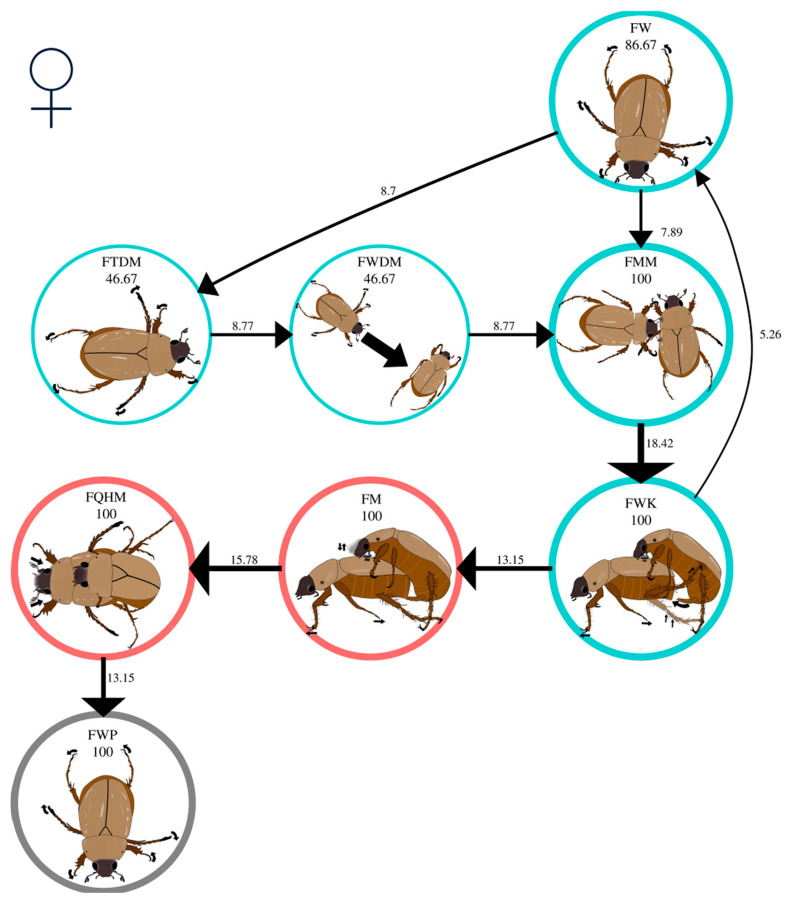
Mating behavior of once-mated *Cyclocephala barrerai* females. The thickness of the line is proportional to the percentage value of transitions; this value is to the right or above the line. The line thickness of the circles is proportional to the percentage of the behavior; the percentage value is below the behavior name. The circle’s color represents the phases of the mating process: blue = premating; red = mating; and gray = postmating.

**Table 1 insects-16-00613-t001:** Repertoire of the *Cyclocephala barrerai* males’ mating behavior.

Abbreviation of the Elements of the Act	Name of the Elements of the Act	Description
MW	Male walks	The male walks along the periphery of the arena without any apparent direction or orientation.
MQ	Male is quiet	The male is still with his head down.
MHAM	Male head movement and antennation	The male moves his antennae up and down, accompanied by head movements and alternating contraction and extension of the antennae.
MTDF	Male turns in direction of female	The male turns his head and thorax in the direction the female is walking.
MWDF	Male walks in direction of female	The male, wandering, changes course and walks directly toward the female.
MATFB	Male antennation on female’s body	The male touches the lateral area of the thorax or abdomen of the female with his antennae.
MHFAR	Male holds female and antennal rubbing	The male holds the female’s elytra with his tarsi and rubs his antennae on her elytra.
MEE	Male extends aedeagus	The male folds his abdomen and exposes the aedeagus.
MIE	Male introduces aedeagus	The male inserts the aedeagus into the female’s pygidium.
MM	Male mounting the female	The male mounts the female, holding her with the tarsal claws of his forelegs. The male inserts the aedeagus into the female’s pygidium, rubs its antennae on her elytra, and occasionally moves its abdomen.
MG	Male guarding the female	The male is positioned on top of the female, holding her with the tarsal claws of his forelegs. The aedeagus is neither inserted nor exposed, and no antennal rubs are observed.
MWP	Male walking (postcopulatory)	The male releases the female and walks away.
MQP	Male quiet (postcopulatory)	The male is still with his head down. Only in remating behavior.

**Table 2 insects-16-00613-t002:** Significant first-order behavior transitions in virgin *Cyclocephala barrerai* males.

		MW	MHAM	MTDF	MWDF	MATFB	MHFAR	MEE	MIE	MM	MG	MWP	Σ
MW	Obs		9 *	20 *									29
	Exp		1	3.14									
	χ^2^		64	90.52									
	T		3.48	7.75									
MHAM	Obs	5 *		8 *									13
	Exp	0.7		1.41									
	χ^2^	26.14		30.8									
	T	1.93		3.1									
MTDF	Obs				30 *								30
	Exp				3.48								
	χ^2^				202.1								
	T				11.62								
MWDF	Obs					30 *							30
	Exp					3.95							
	χ^2^					171.7							
	T					11.62							
MATFB	Obs						34 *						34
	Exp						4.48						
	χ^2^						194.5						
	T						13.17						
MHFAR	Obs	9 *						27 *					36
	Exp	1.95						3.76					
	χ^2^	24.48						143.6					
	T	3.48						10.46					
MEE	Obs								27 *				27
	Exp								3.76				
	χ^2^								143.6				
	T								10.46				
MIE	Obs									27 *			27
	Exp									3.76			
	χ^2^									143.6			
	T									10.46			
MM	Obs										27 *		27
	Exp										3.76		
	χ^2^										143.6		
	T										10.46		
MG	Obs											5 *	5
	Exp											0.09	
	χ^2^											267.8	
	T											1.93	
Σ		14	9	28	30	30	34	27	27	27	27	5	258

Obs = observed value; Exp = expected value; T = transition; * = statistical significance.

**Table 3 insects-16-00613-t003:** Significant first-order behavior transitions in once-mated *Cyclocephala barrerai* males.

		MQ	MW	MHAM	MTDF	MWDF	MATFB	MHFAR	MEE	MIE	MM	MG	MWP	MQP	Σ
MQ	Obs		8 *	4 *											12
	Exp		0.96	0.64											
	χ^2^		51.6	17.6											
	T		5.3	2.7											
MW	Obs	7 *		4 *	10 *		5 *								26
	Exp	1.22		1.39	2.44		3.31								
	χ^2^	27.3		4.9	23.4		0.86								
	T	4.7		2.6	6.7		3.35								
MHAM	Obs				4 *										4
	Exp				0.37										
	χ^2^				35.6										
	T				2.68										
MTDF	Obs					14 *									14
	Exp					1.31									
	χ^2^					123									
	T					9.4									
MWDF	Obs						14 *								14
	Exp						1.78								
	χ^2^						83.8								
	T						9.4								
MATFB	Obs							10 *							10
	Exp							0.67							
	χ^2^							130							
	T							6.7							
MHFAR	Obs		4 *						14 *						18
	Exp		1.45						1.69						
	χ^2^		4.4						89.6						
	T		2.68						9.4						
MEE	Obs									14 *					14
	Exp									1.31					
	χ^2^									123					
	T									9.4					
MIE	Obs										14 *				14
	Exp										1.31				
	χ^2^										123				
	T										9.4				
MM	Obs											14 *			14
	Exp											1.31			
	χ^2^											123			
	T											9.4			
MG	Obs												5 *	4 *	9
	Exp												0.3	0.24	
	χ^2^												73.6	58.9	
	T												3.3	2.7	
Σ		7	12	8	14	14	19	10	14	14	14	14	5	4	149

Obs = observed value; Exp = expected value; T = transition; * = significant.

**Table 4 insects-16-00613-t004:** Elements of the *Cyclocephala barrerai* females’ mating behavior.

Abbreviation of the Elements of the Act	Name of the Elements of the Act	Description
FW	The female walks.	The female walks on the periphery of the arena without apparent direction or orientation.
FQ	The female is quiet.	The female is still with her head down.
FHMCA	The female moves her head and antennae.	The female moves her head up and down and contracts and extends her antennae.
FTDM	The female turns in the direction of the male.	The female turns her head and thorax towards the male.
FWDM	The female walks in the direction of the male.	The female walks straight toward the male, occasionally extending her antennae.
FMA	The female moves away from the male.	The female stops walking on the periphery of the arena or has no apparent orientation and moves away from the male.
FMM	The female meets the male.	The male and female meet and interact.
FWK	The female walks forward while kicking the male.	As the male attempts to grasp the lateral areas of the female’s elytra, the female moves forward while continuously thrusting back one of the second or third pair of legs, slamming it against the male and pushing him backward.
FM	The female is in mounting.	The female walks with the male on top. The male has inserted his aedeagus into the female’s pygidium and moves his head and antennae.
FQHM	The female is quiet and has head movements.	The female stops and moves her head up and down, stretching and contracting her antennae.
FWP	The female walks (postcopulatory).	The female walks with or without the male on top of her, and the male’s aedeagus is not inserted.

**Table 5 insects-16-00613-t005:** Significant first-order behavior transition in virgin *Cyclocephala barrerai* females.

		FW	FHC	FMA	FTDM	FWDM	FMM	FWK	FM	FQHM	FWP	Σ
FW	Obs		10 *	5 *	7 *							35
	Exp		2.3	1.97	2.62							
	χ^2^		25.77	4.66	7.32							
	T		4.69	2.34	3.28							
FQ	Obs		4 *									4
	Exp		0.26									
	χ^2^		53.12									
	T		1.87									
FHMCA	Obs			7 *	4 *							11
	Exp			0.61	0.82							
	χ^2^			66.93	12.33							
	T			3.28	1.87							
FMA	Obs	4 *			5 *							9
	Exp	0.54			0.67							
	χ^2^	22.17			27.98							
	T	1.87			2.34							
FTDM	Obs					17 *						17
	Exp					1.35						
	χ^2^					181.4						
	T					7.98						
FWDM	Obs						16 *					16
	Exp						2.17					
	χ^2^						88.14					
	T						7.51					
FMM	Obs							34 *				34
	Exp							5.42				
	χ^2^							150.7				
	T							15.96				
FWK	Obs	9 *							26 *			35
	Exp	2.13							4.27			
	χ^2^	22.15							110.5			
	T	4.22							12.2			
FM	Obs									26 *		26
	Exp									3.17		
	χ^2^									164.41		
	T									12.2		
FQHM	Obs										26 *	26
	Exp										3.17	
	χ^2^										164.4	
	T										12.2	
Σ		13	14	12	16	17	29	34	26	26	26	213

Obs = observed value; Exp = expected value; T = transition; * = significant.

**Table 6 insects-16-00613-t006:** Significant first-order behavior transition in once-mated female *Cyclocephala barrerai*.

		FW	FTDM	FWDM	FMM	FWK	FM	FQHM	FWP	Σ
FW	Obs		10 *		9 *					19
	Exp		1.6		3.16					
	χ^2^		41.6		10.7					
	T		8.7		7.9					
FTDM	Obs			10 *						10
	Exp			0.87						
	χ^2^			94.8						
	T			8.77						
FWDM	Obs				10 *					10
	Exp				1.66					
	χ^2^				41.6					
	T				8.7					
FMM	Obs					21 *				21
	Exp					3.86				
	χ^2^					75.8				
	T					18.4				
FWK	Obs	6 *					15 *			21
	Exp	1.1					2.76			
	χ^2^	21.6					54.1			
	T	5.26					13.15			
FM	Obs							18 *		18
	Exp							2.84		
	χ^2^							80.8		
	T							15.7		
FQHM	Obs								15 *	15
	Exp								1.97	
	χ^2^								85.9	
	T								13.15	
Σ		6	10	10	19	21	15	18	15	114

Obs = observed value; Exp = expected value; T = transition; * = significant.

## Data Availability

The original contributions presented in this study are included in the article/Appendix A. Further inquiries can be directed to the corresponding author.

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
