# Peer review of "Mating Behavior of Cyclocephala barrerai Martínez (Coleoptera: Melolonthidae)"

_insects, 2025, doi:10.3390/insects16060613_

Round 1
Reviewer 1 Report
Comments and Suggestions for Authors
Please see the attachment.

Author Response
"Please see the attachment."

Reviewer 2 Report
Comments and Suggestions for Authors
An interesting work, with a lot of detail, nicely illustrated. Therefore, the review listed a fair number of minor omissions. Please, consider them patiently, particularly in tables.
Good luck!

Author Response
"Please see the attachment."

Reviewer 3 Report
Comments and Suggestions for Authors
The manuscript insects-3618718 offers new and original insights into the mating behavior of Cyclocephala barrerai. However, some issues regarding the overall structure and clarity of the manuscript need to be addressed:
- Overall Structure: The reading is generally hard; ideas within and among paragraphs sometimes lacks fluidity.
- Introduction Section: Including information more detailed information about the economic or ecological importance of the species would improve the study justification. Please include specific objectives to improve the chronological and logical order among sections. The methodology, results and discussion sections should be clearly aligned with each objective of the study (which are also not clearly stated). Each objective should be addressed systematically across these sections to ensure coherence and to help the reader follow the progression of the research. Reorganizing the content in this way would significantly improve the clarity and impact of the manuscript.
- Figures and tables: Figure and table captions should be self-contained and understandable without reference to the main text. They must include essential information to clearly describe the content they present.
Note: Specific comments and detailed suggestions have been provided in the attached document.

Please ensure that the entire document is reviewed for spelling and the flow of ideas, as mentioned in the Comments and Suggestions for Authors.
Author Response
"Please see the attachment."

Round 2
Reviewer 1 Report
Comments and Suggestions for Authors
- The introduction is too redundant and the topic is not focused, including Line 45-85.
- There are lack of essential infromation about the once-mated in the section of introduction.
- What's mean of the black block in the lower left corner in Figure 1.
- What's the role of L.495-500 in the section of discussion.
- There are lack of discussion in depth for the difference of mating behaviors of both virgin and once-mated female.
Round 3
Reviewer 1 Report
Comments and Suggestions for Authors
I am satisfied with this revision.